# $^{39}$Ar dating with small samples provides new key constraints on ocean ventilation

Sven Ebser [1], Arne Kersting[2], Tim Stöven [3], Zhongyi Feng [1], Lisa Ringena[1], Maximilian Schmidt [1,2], Toste Tanhua [3], Werner Aeschbach [2,4] & Markus K. Oberthaler [1]

Ocean ventilation is the integrated effect of various processes that exchange surface properties with the ocean interior and is essential for oxygen supply, storage of anthropogenic carbon and the heat budget of the ocean, for instance. Current observational methods utilise transient tracers, e.g. tritium, $SF_6$, CFCs and $^{14}$C. However, their dating ranges are not ideal to resolve the centennial-dynamics of the deep ocean, a gap filled by the noble gas isotope $^{39}$Ar with a half-life of 269 years. Its broad application has been hindered by its very low abundance, requiring 1000 L of water for dating. Here we show successful $^{39}$Ar dating with 5 L of water based on the atom-optical technique Atom Trap Trace Analysis. Our data reveal previously not quantifiable ventilation patterns in the Tropical Atlantic, where we find that advection is more important for the ventilation of the intermediate depth range than previously assumed. Now, the demonstrated analytical capabilities allow for a global collection of $^{39}$Ar data, which will have significant impact on our ability to quantify ocean ventilation.

[1] Kirchhoff-Institute for Physics, Heidelberg University, 69120 Heidelberg, Germany. [2] Institute of Environmental Physics, Heidelberg University, 69120 Heidelberg, Germany. [3] GEOMAR Helmholtz Centre for Ocean Research Kiel, 24148 Kiel, Germany. [4] Heidelberg Center for the Environment (HCE), Heidelberg University, 69120 Heidelberg, Germany. Correspondence and requests for materials should be addressed to S.E. (email: oceanArTTA@matterwave.de)

The well-mixed surface layer of the ocean exchanges properties with the atmosphere through air–sea gas exchange. Various processes such as advection and eddy diffusion are responsible for transporting surface waters with their corresponding properties to the ocean interior. The integrated effect of such water mass exchange is termed ocean ventilation. Knowledge of its temporal and spatial variations is essential for a reliable prediction of the Earth system's response to climate change[1,2]. Thus, systematic observation on a global scale is desired. With the new capabilities reported here, this appears to be feasible now.

Quantifying ventilation includes an estimation of the time since the water was last in contact with the atmosphere, that is, the age of the water. It can be accessed by observations of transient tracers which encode time information via radioactive decay or a time-dependent input function[3]. The short atmospheric histories of well-established transient tracers such as chlorofluorocarbons (CFCs) and sulfur hexafluoride ($SF_6$)[4] cover only the past 70 years, preventing the dating of the slowly ventilated part of the global ocean. There, time-scales of approximately 1000 years are estimated from $^{14}C$ measurements[5]. However, these results are rather uncertain due to the complex carbon dynamics, long air–sea equilibration time and long half-life compared to ocean ventilation. The chemically inert noble gas isotope $^{39}Ar$ with a half-life of 269 years has long been identified as the ideal tracer for the time-scales of deep ocean circulation[6–10]. However, even though argon is a common gas in the atmosphere, the desired $^{39}Ar$ isotope is extremely rare, due to its very low isotopic abundance of $8 \times 10^{-16}$. The corresponding low activity necessitates samples of ~1000 L of water for $^{39}Ar$ detection by low-level counting (LLC) of the radioactive decays[6,10,11]. This large sample size hinders routine measurements of ocean samples. Here we show that our method for analysing $^{39}Ar$, which we call argon trap trace analysis (ArTTA), reduces the required water volume to 5 L. This makes large-scale ocean surveys feasible as taking 5 L of water can be readily integrated into standard water sampling procedures on research vessels.

## Results and Discussion

**Argon trap trace analysis**. For the detection of long-lived radioisotopes, it is more efficient to count atoms rather than radioactive decays[12]. For example, atom counting by accelerator mass spectrometry (AMS) dramatically reduced sample-size requirements for $^{14}C$ dating. AMS is not easily applicable for noble gases, yet atom counting and the related substantial reduction of sample size becomes possible by employing the modern atom-optical technique known as atom trap trace analysis (ATTA). It utilises techniques from the field of atomic physics to detect rare isotopes down to the $10^{-16}$ level. It exploits shifts of the optical resonance frequency due to different isotopic mass and nuclear spin. The high background of abundant isotopes hinders selection by a single resonant excitation, but the sensitivity is strongly enhanced by many cycles of photon absorption and subsequent spontaneous emission. Therefore, ATTA's outstanding isotopic selectivity is based on millions of resonant photon scattering events required for cooling, trapping and detecting single atoms inside a magneto-optical trap (MOT).

The general concept of ATTA has first been demonstrated for the rare isotopes $^{85}Kr$ (half-life of 10.76 years) and $^{81}Kr$ (half-life of 229,000 years)[13,14] and is applied for dating groundwater[15–17] and ice[18]. While the first $^{39}Ar$ detection by this approach was reported in a proof of concept experiment in 2011[19], the first explicit demonstration for dating groundwater samples was achieved in 2014[20]. The first apparatus used in that study still required ~1000 L of water. Since then the setup has been significantly improved by doubling the count rate, ensuring

reliability and employing well-characterised enriched reference samples. The crucial step for the reduction of the necessary sample size is the implementation of gas recirculation in an optimised vacuum system. With this ArTTA system, $^{39}Ar$ quantification is now possible with only 2 mL STP (standard temperature and pressure) of argon, which can be extracted from 5 L of water. Thus, a complete ocean depth profile of $^{39}Ar$ can be sampled from one standard hydrographic cast equipped with 10 L Niskin sampling bottles, fulfilling the requirements for broader application of $^{39}Ar$ in oceanography[21].

**$^{39}Ar$ depth profiles**. Here we apply our new analytical capabilities to explore the ventilation regime in the Eastern Tropical North Atlantic, in the context of investigations of the Oxygen Minimum Zone in this region. In 2015, three depth profiles were taken with one single hydrographic cast per profile during the research cruise M116 on research vessel Meteor. Two profiles (#44, #55) originate from the centre of the Oxygen Minimum Zone and one profile (#82) from the Cape Verde Ocean Observatory (http://cvoo.geomar.de/) (Fig. 1). The ocean water was sampled by closing three 10-L Niskin bottles of a 24 bottle rosette per depth and transferring ~7 L of the content of each Niskin bottle into an evacuated 27 L commercial propane gas bottle. By combining three Niskin bottles to one sample of ~20 L, in total 24 sampling containers from eight different depths at three sampling sites were taken. Gas extraction and argon separation from these ~20 L samples resulted in a total argon yield between 5 and 8 mL STP, consistent with the temperature-dependent solubility[22]. Each sample was analysed by at least two independent measurements, where the analytical uncertainties are dominated by counting statistics. As seen in Fig. 1, we find a clear decline in $^{39}Ar/Ar$ ratio in the upper 1000 m, a minimum at ~3000 m and a slight increase towards the bottom at 4000 m. For comparison, we also include the three historic $^{39}Ar$ samples closest to our sampling positions, which were taken in 1981 and analysed by LLC[11]. The observed differences in $^{39}Ar/Ar$ ratios are consistent with the known meridional gradients in tracer concentrations characteristic for this region. Note that these historic $^{39}Ar$ data are integrated over a depth interval of 600 to 800 m as indicated by the vertical error bars due to the required large sample size of 1000 L. The ArTTA technique allows for 24 $^{39}Ar$ samples during one single cast (with a 24 bottle rosette) that takes ~3 h of ship-time, which is about the same amount of time needed for one large volume sample required for LLC.

**Transit time distribution**. In the following we discuss how the obtained $^{39}Ar/Ar$ ratios constrain the ventilation in this area. Since ventilation always implies mixing along a multitude of advective and diffusive paths, it is necessary to describe a water sample by a distribution of transit times $\tau$ rather than one distinct age. This age distribution is known as transit time distribution (TTD)[23,24]. Given the TTD $G(\tau, \mathbf{r})$ for a location $\mathbf{r}$ in the ocean interior and a time-independent flow, the concentration at sampling time $t_s$ is described by

$$c(t_s, \mathbf{r}) = \int_0^\infty c_0(t_s - \tau) \cdot e^{-\lambda\tau} \cdot G(\tau, \mathbf{r}) d\tau, \qquad (1)$$

where $c_0(t_s - \tau)$ corresponds to the surface concentration during source year $t_s - \tau$ in the dominant surface source region of the water sample. The exponential term accounts for the decay rate of radioactive transient tracers. $G(\tau, \mathbf{r})$ is the Green's function of the flow model for the given location[24]. One established method to deconvolute the TTD is to use a maximum entropy approach, where the value of $^{39}Ar$ data has been clearly demonstrated[9]. For illustration of the utility of $^{39}Ar$ observations, we use the inverse

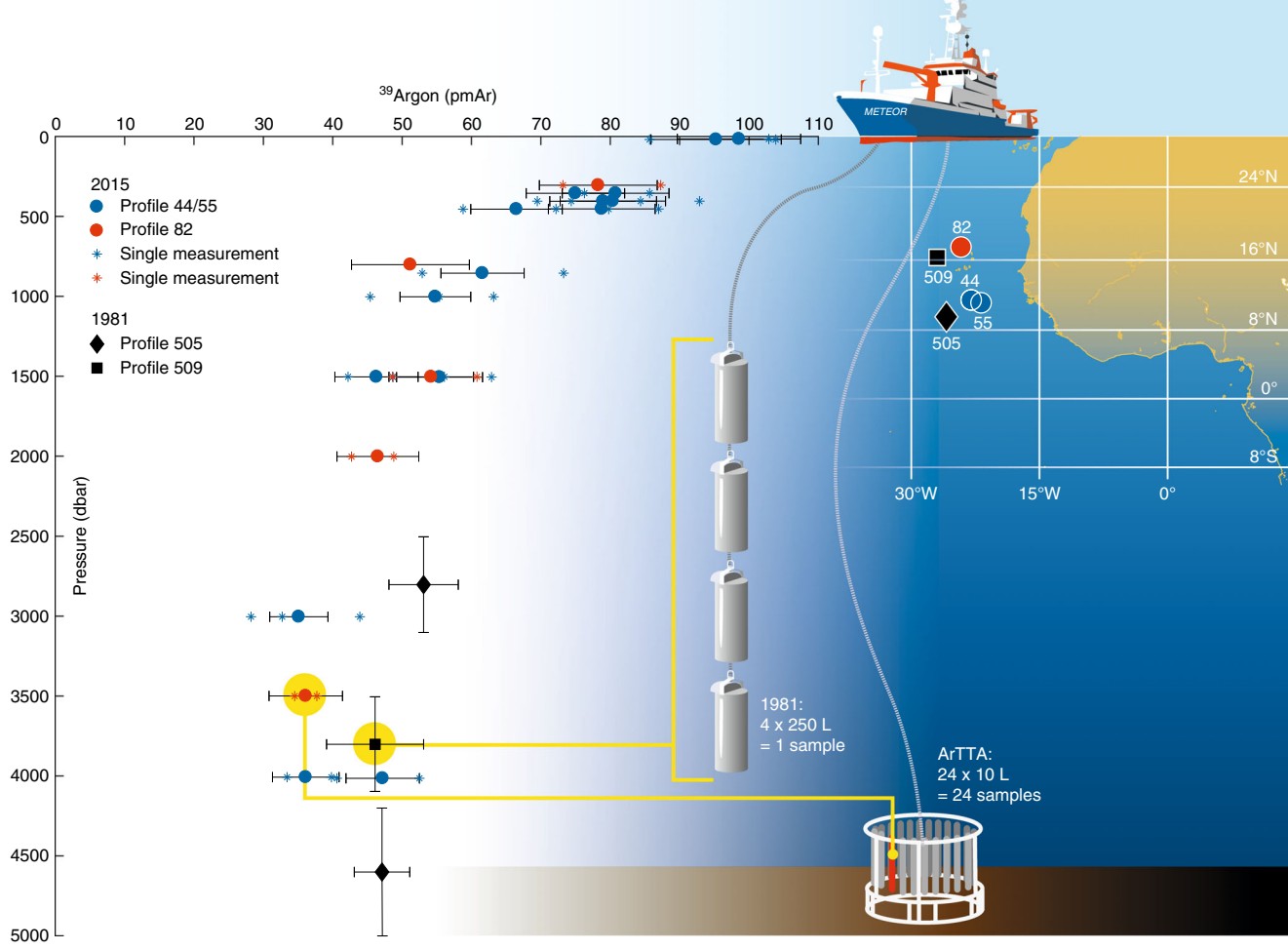

**Fig. 1** $^{39}$Ar/Ar ratios in percent of modern Argon (pmAr) for different depths in the Eastern Tropical North Atlantic (blue and red dots). Each sample was analysed with at least two independent measurements and the horizontal error bars represent the uncertainties of the combined single measurements (single stars). The sensitivity and efficiency of ArTTA allows for a full-depth profile from just one CTD cast corresponding to sample sizes as small as 5 L of water. For comparison, three historic (1981) data points[11] from two specific sites are included (black square and rhombi); the required 1000 L for LLC analysis was obtained by combining four 250 L sampling bottles integrating over a depth range of 600 to 800 m (vertical error bars). The map shows the western part of Africa and the sampling stations of this study and those from 1981 around the archipelago of Cape Verde

Gaussian function (IG-TTD)[25], which is defined by its first two moments: the mean of the distribution $\Gamma$ and the width of the distribution $\Delta$.

$$G(\tau) = \sqrt{\frac{\Gamma^3}{4\pi\Delta^2\tau^3}}\exp\left(\frac{-\Gamma(\tau-\Gamma)^2}{4\Delta^2\tau}\right). \quad (2)$$

This TTD is the analytical solution for a one-dimensional flow model with time-invariant advective velocity, diffusivity and one single source region. These assumptions are reasonable for atmospheric and ocean transport studies and possible limitations are discussed in the Methods. The $\Delta/\Gamma$ ratio indicates the eddy-diffusive compared to the advective transport characteristics of a water parcel; the lower $\Delta/\Gamma$, the more dominant the advection. A $\Delta/\Gamma$ ratio of 0.4–0.8 indicates advectively dominated transport, whereas a high ratio of 1.2–1.8 indicates transport dominated by diffusive processes. Several approaches have been proposed for constraining the parameters of the IG-TTD based on tracer data; here we use the method outlined and thoroughly discussed by Stöven and Tanhua[26]. This approach is based on constraining the

TTD for each sample individually. The two parameters of the IG-TTD can be constrained using observed concentrations of at least two transient tracers with sufficiently different time information covering the expected transit time range of the water parcel[3,25].

As the second independent tracer we choose CFC-12, which was sampled at the same, or nearby, position. In Fig. 2a the corresponding $^{39}$Ar/Ar ratios and CFC-12 concentrations are depicted and are well above the limit of quantification. The uncertainties of the $^{39}$Ar/Ar ratios are limited by the counting statistics and for CFC-12 by the system performance of the on-board measurements and the known systematic uncertainties. In Fig. 2b we illustrate for three different depths how both measured tracers constrain possible parameter combinations ($\Gamma$, $\Delta/\Gamma$). Considering the year of sampling and the input function of the tracers, a particular tracer concentration can correspond to a range of combinations of $\Delta$ and $\Gamma$, which we plot as $\Delta/\Gamma$ vs. $\Gamma$[26]. For two (or more) independent tracers multiple such curves can be analysed and the range of possible combinations of $\Delta$ and $\Gamma$ can be constrained, as indicated by the intersecting areas. The precision of the results is limited by the combination of the uncertainty of the input function (including saturation of the tracer at water mass formation), the analytical and interpolation

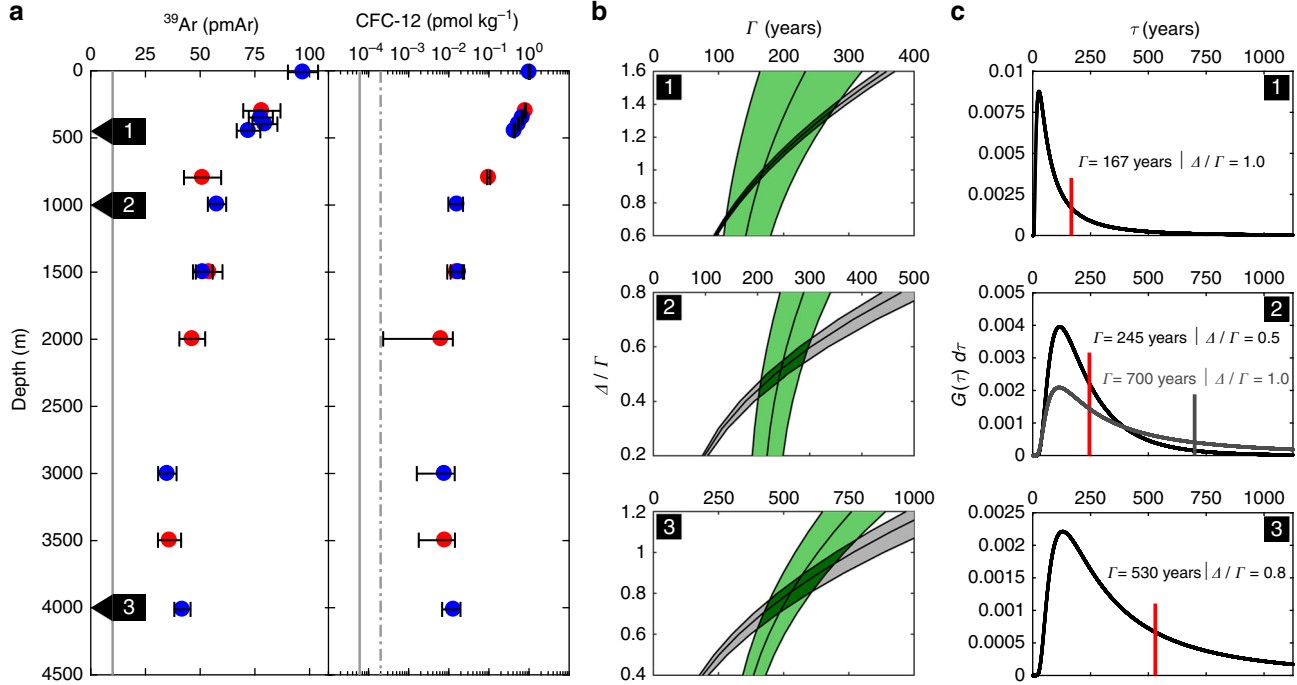

**Fig. 2** Constraining inverse Gaussian transit time distributions (IG-TTDs). **a** shows the $^{39}$Ar/Ar ratio profiles on a linear and the CFC-12 concentration profiles on a logarithmic scale: red markers for profile #82 and blue markers for the combined profile #44/#55. The indicated uncertainties for $^{39}$Ar are given by the counting statistics, and for CFC-12 due to the on-board system performance and the known systematic uncertainties. The solid and dashed grey vertical lines indicate the limits of detection and quantification, respectively. **b** illustrates how $^{39}$Ar (light green) and CFC-12 (grey) data constrain possible parameter combinations ($\Gamma$, $\Delta/\Gamma$) of the IG-TTD given by their intersecting dark green area. **c** shows the corresponding distributions. For comparison we added in the middle panel the distribution obtained using only CFC-12 data with the commonly assumed $\Delta/\Gamma$ unity ratio

uncertainties, as well as by the assumption of the propagator $G$ having a simple IG transit time dependence. For our study, we assume for $^{39}$Ar a constant input of 100 pmAr and for CFC-12 reported data for the northern hemisphere[27]. In Fig. 2c the corresponding IG-TTDs are depicted.

**Implications for ventilation regimes**. The water column can be tied to different ventilation regimes as described by $\Delta/\Gamma$ ratios and mean ages $\Gamma$ (Fig. 3a, b) or by the dominant water masses as illustrated by their salinity and temperature characteristics (Fig. 3c). The water samples above ~800 m correspond to the Atlantic Central Waters. Here we find a unity ratio $\Delta/\Gamma = 1$ and mean ages consistent with estimates based on the tracer couple SF$_6$/CFC-12 for this ocean region[28]. It is important to note that this regime is only poorly constrained by $^{39}$Ar, which does not resolve small absolute age differences important in the young age regime due to the larger analytical uncertainty in comparison to other tracers such as SF$_6$. For deeper and thus older water, the SF$_6$/CFC-12 tracer couple is not applicable for determining the age distribution, but $^{39}$Ar/CFC-12 gives new insights. For example, in intermediate depths (1000 m –2000 m), where Antarctic Intermediate Water (AAIW) and Labrador Sea Water (LSW) dominate, we find $\Delta/\Gamma = 0.5 - 0.6$ based on our data. This is in stark contrast to the $\Delta/\Gamma$ unity ratio, which has been commonly applied to this depth interval before[28–30]. Our findings of these low $\Delta/\Gamma$ ratios reveal that the ventilation there is of a more advective nature than previously assumed. Additionally, the first reliable mean ages $\Gamma$ in this deeper region can be derived due to the unique half-life of $^{39}$Ar and increase with depth from 200 to 400 years (Fig. 3b). The deepest ventilation regime with $\Delta/\Gamma \sim 0.9$ is found for water mainly composed of North Atlantic Deep Water (NADW) and Antarctic Bottom Water (AABW). We identify an increase in mean age up to $\Gamma = 800$ years at 3000 m

depth, followed by a decrease towards the ocean floor due to the better ventilated AABW. Although the presence of AABW in this region is well known, the faster ventilation of the AABW in comparison to the NADW has previously not been described and is now verified by the $^{39}$Ar data from this region. In these areas, the TTD can only be poorly constrained due to low CFC-12 signals, resulting in large uncertainty of the inferred mean age $\Gamma = 810^{+1200}_{-320}$ years for our oldest sample at 3000 m depth. Alternatively, using the extracted mean $\Delta/\Gamma \sim 0.9$ for the third ventilation regime we find $\Gamma = 754^{+138}_{-115}$ years based on our $^{39}$Ar data. Thus, $^{39}$Ar provides essential information on ventilation ages for old waters. Results from transient tracer observations in the same region using the maximum entropy method[31] indicate slightly lower mean ages than reported here, although within the uncertainties of the two methods.

**Broader implications and prospects**. Once the TTD of a water body is known, one can derive concentrations of substances which have not been or which cannot be measured directly, such as anthropogenic carbon ($C_{ant}$), as long as their input functions are known. We estimate $C_{ant}$ by applying Eq. (1) to the TTD as deduced from transient tracers, using the well-known input function of $C_{ant}$. In the studied area, we find the $C_{ant}$ concentration within the AAIW/LSW range for profile #44/#55 at 1000 m to be ~40% higher using the assumption of an IG-TTD with $\Delta/\Gamma$ deduced from our $^{39}$Ar data (i.e. 0.5–0.6), than obtained using the commonly assumed $\Delta/\Gamma$ unity ratio (Fig. 4). For the water column between 1000 m and 2000 m in ventilation regime II, this difference adds up to 2–3 mol m$^{-2}$ which should be compared to the total column inventory of ~25 mol m$^{-2}$ [30].

The presented work is an example how applied quantum technology developed in the context of fundamental research in atomic physics contributes to the advance of other fields, such as

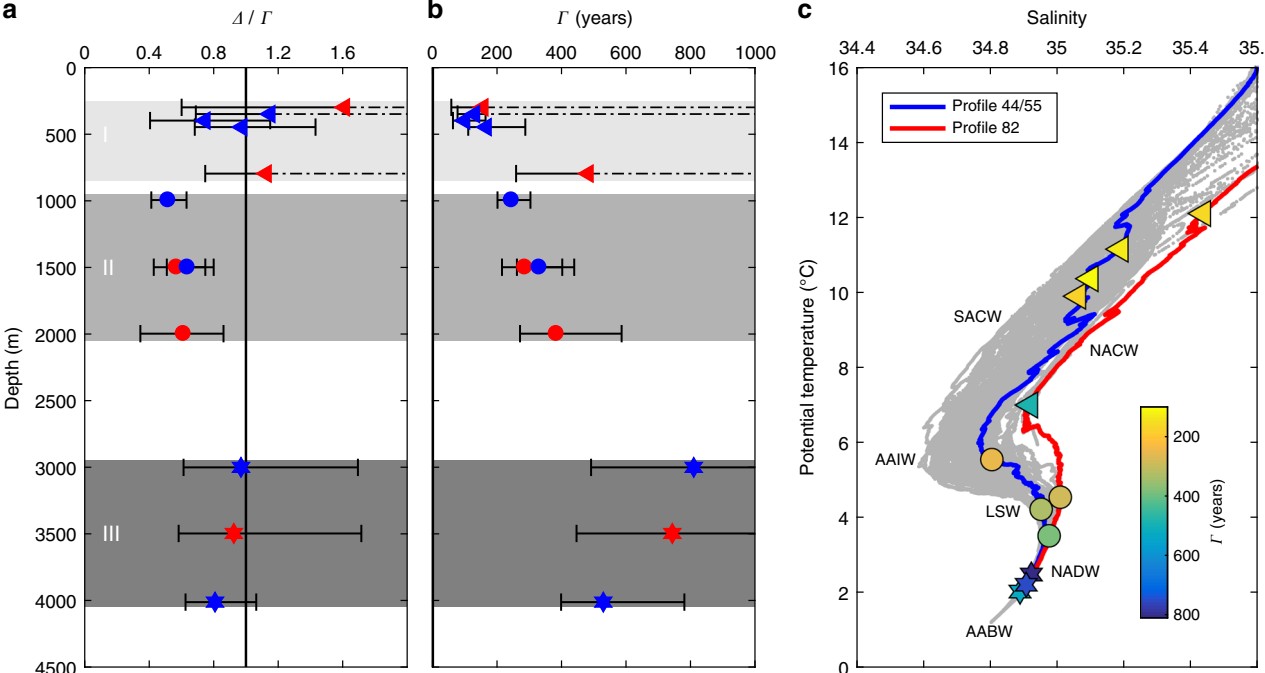

**Fig. 3** Identified ventilation regimes for the Eastern Tropical North Atlantic. Estimated $\Delta/\Gamma$ ratios (**a**) and mean ages $\Gamma$ (**b**). We identify three different ventilation regimes as indicated by the grey areas, characterised by $\Delta/\Gamma \approx 1$ above 800 m (regime I) and below 3000 m (regime III) and $\Delta/\Gamma \approx 0.5$ in the intermediate regime (regime II). **c** Temperature vs. salinity of the study area: the grey dots represent the temperature and salinity data of the whole cruise, while the blue and red lines highlight the S/T curves for profiles #44/#55 and #82, respectively. The colour code represents the mean age $\Gamma$ of the analysed samples. AABW: Antarctic Bottom Water, NADW: North Atlantic Deep Water, LSW: Labrador Sea Water, AAIW: Antarctic Intermediate Water, SACW: South Atlantic Central Water, NACW: North Atlantic Central Water

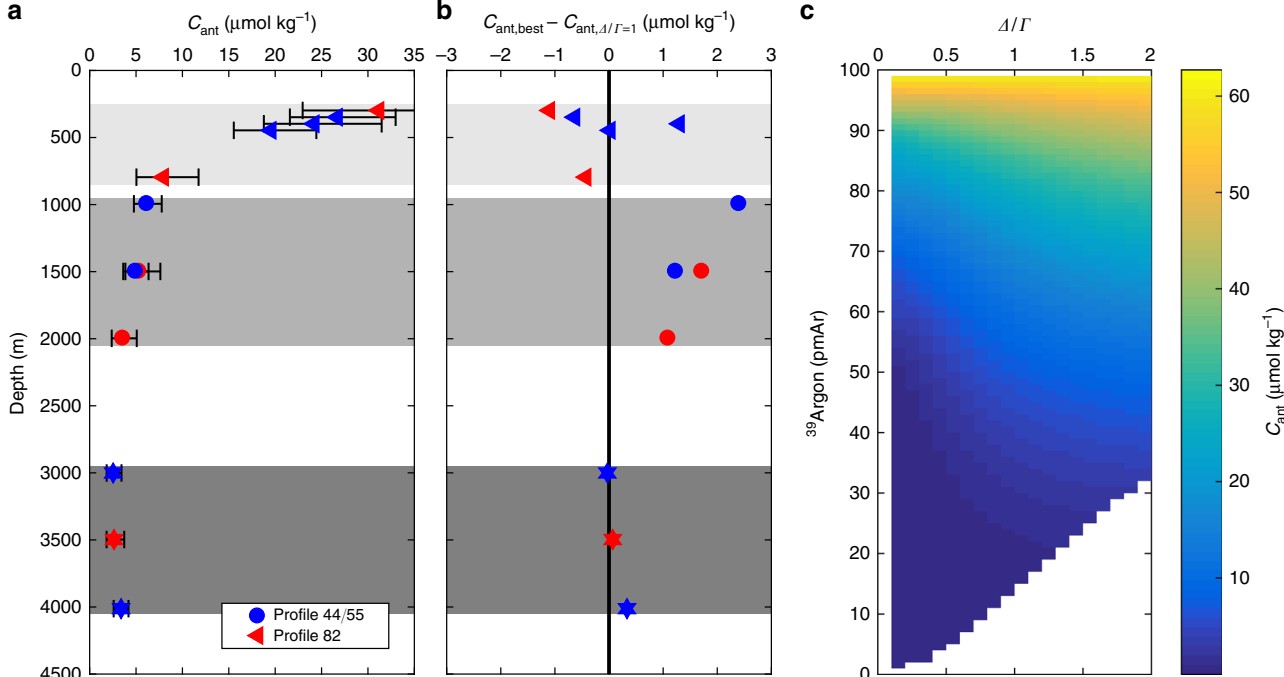

**Fig. 4** Estimated anthropogenic carbon content for all ventilation regimes. **a** shows the anthropogenic carbon concentration based on the constrained IG-TTD. The uncertainties are calculated assuming the best estimate for $\Delta/\Gamma$ and taking the uncertainty range for $^{39}$Ar into account. The error bars are small in the deep regime due to the fact that for a mean age much older than the beginning of significant anthropogenic carbon production variations of the TTD have no large effect on the anthropogenic carbon concentration. **b** shows the difference in $C_{ant}$ concentration of the constrained IG-TTD and the standard parameterisation of $\Delta/\Gamma = 1$. **c** shows the dependence of $^{39}$Ar and $C_{ant}$ concentration for different $\Delta/\Gamma$. The calculations are based on a salinity of 35, a potential temperature of 10 °C and a maximum mean age of 2500 years

oceanography. The demonstrated ArTTA method for small sample sizes makes the unique time-scale of $^{39}$Ar accessible for large-scale ocean surveys. This can increase our knowledge of ocean dynamics significantly and, with that, support ocean and climate modelling. There are many other areas in environmental sciences, such as glaciology, limnology and groundwater research, where the novel analytical capabilities and much smaller sample sizes will enable new applications and lead to new insights.

## Methods

**ATTA for argon with small sample sizes**. The basic concept of our atomic beam apparatus for argon follows the established route[14,19,20] with some modifications as mentioned below. It consists of a radio frequency discharge source, which prepares the atoms in the metastable state necessary for laser cooling. The source is cooled with liquid nitrogen to reduce the initial velocity spread of the atomic beam. The divergent atomic beam, with a longitudinal mean velocity of 270 m s$^{-1}$, is collimated with transverse laser cooling in a tilted mirror setup and focused by a magneto-optical lens. Subsequently, the atoms with a maximum velocity of 600 m s$^{-1}$ are longitudinally slowed down in a 1.8 m long increasing field Zeeman slower. The final velocity of the atoms leaving the Zeeman slower is around 70 m s$^{-1}$ and is chosen above the capture velocity of the MOT. Thus, the atomic beam is prevented to diverge rapidly at the end of the Zeeman slower. An additional laser frequency builds a second longitudinal slowing stage together with the rising slope of the magnetic field of the MOT. There the atoms are refocused by the MOT and losses due to the divergence of the atomic beam are reduced significantly. Finally, the atoms are trapped inside the MOT and detected by their fluorescence with an avalanche photodiode with a high time resolution of 1 ms binning and spatially resolved on a charge-coupled device camera. The detection threshold is set so that the rate of falsely counted atoms due to background noise is kept below 1 atom in 100 h. The corresponding detection efficiency is 94% of all trapped $^{39}$Ar atoms. The loss of 6% results mainly from atoms which are trapped shorter than 40 ms and thus are difficult to be detected unambiguously.

An additional 802 nm laser de-excites isotope selectively 99% of all metastable $^{40}$Ar atoms to the ground state, which are responsible for half of the background light on the single atom detection. The photons (843 nm) emitted by the de-exciting $^{40}$Ar atoms are detected by a photodiode which allows the monitoring of the atomic beam flux. Three differential pumping tubes together with seven turbo molecular pumps build up a pressure gradient from $4.8 \times 10^{-6}$ mbar in the source chamber down to $1.4 \times 10^{-7}$ mbar inside the MOT, which results in a mean lifetime of single $^{39}$Ar atoms of 290 ms. Thereby, the argon throughput of the source is ~50 mL STP h$^{-1}$.

By closing the vacuum completely (i.e. the argon sample recirculates in the system and no roughing pump removes gas from the system) the required sample size can be reduced by more than two orders of magnitudes. In this so-called recycling or closed mode, the sample circulates permanently inside the system. Outgassing reactive gases, such as nitrogen and water, are removed by a getter pump, which does not affect noble gases. The argon sample will remain in the vacuum chamber and stays clean during the measurement process. A stable operation over the required measurement time is possible with samples sizes >1 mL STP of argon.

For further optimisation, the pressure inside the differential pumping tubes was simulated by a Monte Carlo simulation. By doubling the total pumping speed, we achieve an atmospheric count rate of up to 7.0 atoms h$^{-1}$, which is an improvement by a factor of 2 compared to the count rate reported previously[20].

**Extraction and separation of argon**. The propane gas bottles containing the collected water samples were shipped to the laboratory in Heidelberg where each sample was degassed and purified. The dissolved gas was extracted by shaking the sample container and trapping the gas on a liquid nitrogen-cooled activated charcoal trap. After 15 min more than 95% of the gas is stored on the trap. In a second step, all reactive gases are removed on a 900 °C titanium sponge getter, while the released hydrogen is trapped on a second titanium getter at room temperature. The final gas fraction only consists of noble gases, thus of >98% argon. With a getter capacity of about 8 L for the relevant reactive gases, more than 10 samples of ~50 L of water can be purified before having to replace the getter material.

Purifications of blank samples and helium leak tests are performed regularly and showing that cross-sample contamination and leakage into the vacuum chamber is negligible.

Two ocean samples were purified per day yielding between 5 and 8 mL STP of argon per sample with a purity and extraction efficiency of >98%.

**$^{39}$Ar analysis**. The purified argon samples were analysed with the atom-optical detection technique ArTTA optimised for $^{39}$Ar and small sample sizes as described above. Our 24 h measurement cycle consists of 20 h of measuring an ocean sample, followed by 2 h of referencing with an enriched sample of a well-known $^{39}$Ar/Ar ratio of $9.60^{+0.33}_{-0.31}$ times the atmospheric one. Finally, the system is flushed for 1 h by running the discharge source with krypton to avoid any significant cross-

sample contamination between the reference and the next environmental sample. After 24 h the next sample is analysed. The currently achieved atmospheric $^{39}$Ar count rate of up to 7.0 atoms h$^{-1}$ enables—depending on the sample concentration —up to 150 counts for both the ocean sample and the reference within one day, leading to 10% uncertainty. For analysing the data, both reference measurements before and after counting the ocean sample are taken into account, resulting in an uncertainty of about 7% for the reference. Due to previous optimisation and characterisation of the apparatus with one million times enriched $^{39}$Ar samples more than 5 years ago, there is still a low but detectable contamination present. This contamination together with a potential cross-sample contamination was quantified with $^{39}$Ar-free underground samples originating from $CO_2$ production wells[32]. The measured count rates were corrected for this effect corresponding to about 10 atoms during a 20 h measurement.

We apply a Bayesian approach for the analysis of the $^{39}$Ar measurements. The reported $^{39}$Ar/Ar ratios are the most probable values and the bounds of the one sigma intervals contain most likely 68.3% of the values. The additional uncertainties due to the contamination effect are included in our analysis, but become only dominant for $^{39}$Ar/Ar ratios below 10 pmAr.

**CFC measurements**. Two purge-and-trap gas chromatographic systems were used for the measurements of the transient tracer CFC-12 during the cruise, slightly modified from[33]. The traps for both systems consisted of 100 cm 1/16 in. tubing packed with 70 cm Heysep D kept at temperatures between −60 and −68 °C during trapping. The gas was desorbed by heating the trap to 130 °C and was passed onto the pre-column consisting of 20 cm Porasil C followed by 20 cm Molsieve 5 A in a 1/8 in. stainless-steel column. A 1/8 in. packed main column consisting of 180 cm Carbograph 1AC (60–80 mesh) and a 50 cm Molsieve 5 A post-column provided chromatographic separation. All columns were kept at 50 °C and detection was performed on an Electron Capture Detector.

The water samples for the determination of CFC-12 were collected in 250 mL ground glass syringes, of which an aliquot of about 200 mL was injected to the purge-and-trap system. Standardisation was performed by injecting small volumes of gaseous standard with the CFC-12 calibrated to the SIO98 scale. The precision was determined to 8 fmol kg$^{-1}$ and the limit of detection to 0.06 fmol kg$^{-1}$.

Since we did not take samples for CFCs on the CTD casts where we collected samples for $^{39}$Ar measurements, we use nearby profiles that are comparable. Profile #54 (N 11.00°, W 22.00°) was used to compare the CFC-12 values to the $^{39}$Ar profiles #44 (N 11.55°, W 23.00°) and #55 (N 11.25°, W 22.00°). For profile #82 (N 17.58°, W 24.30°), we used a CTD taken immediately prior on the same position. Since the sampling for $^{39}$Ar and CFC-12 took place on different CTD casts, we interpolated the CFC-12 data vs. density to match the $^{39}$Ar samples.

**Assumptions and limitations of the IG-TTD approach**. Caveats to the TTD method, as described by Eq. (1), include the assumptions of a steady ocean transport and one effective single dominant source region. The real propagator $G(\tau,$ **r**) of the ocean might depend on both the transit time $\tau$ and the source time $t_s - \tau$ and in reality the sample might be a mixture of water ventilated in different regions, which may have slightly different source concentration histories, especially between the northern and southern hemisphere. Additionally the IG-TTD is a unimodal distribution and thus limited to describe only a single dominant advective-diffusive transport pathway. In our study region, we see a mix of water masses from the North and South Atlantic, so that different forms, at least bimodal shapes, of the TTD are conceivable. Here we use the basic IG-TTD approach for illustration of the utility of $^{39}$Ar observations, keeping potential bias of the inferred mean ages $\Gamma$ and widths $\Delta$ in mind. A more complete, but more complicated method to deconvolute the TTD, which has been shown to benefit from $^{39}$Ar data[9], is the maximum entropy approach.

The precision of the estimation of IG-TTD parameters based on tracers is also limited by the uncertainties about their input function and boundary condition at the ocean surface. The tracer $^{39}$Ar is based on the $^{39}$Ar/Ar ratio and thus independent of the solubility of argon in sea water as well as possible Ar saturation anomalies. In contrast, for chemical tracers such as CFC-12 their solubility equilibrium at the ocean surface during water mass formation needs to be calculated from salinity and potential temperature, which are treated as conservative properties. More importantly, both CFC-12 and $^{39}$Ar/Ar require an estimation of the deviation from full equilibration with the atmosphere at the ocean surface during water mass formation. In areas of deep water formation, deviations from 100% equilibration have been noted, but are neglected in our analysis for simplicity.

## Data availability
The complete data set from the Meteor 116 cruise including the $^{39}$Ar data can be found at PANGAEA: https://doi.org/10.1594/PANGAEA.894708 and https://doi.pangaea.de/10.1594/PANGAEA.886191. The $^{39}$Ar and CFC-12 data of the three profiles analysed in this study can also be found in Supplementary Table 1 and the obtained $\Delta/\Gamma$ ratios, mean ages $\Gamma$ and $C_{ant}$ concentration in Supplementary Table 2. Figure 1 contains the raw $^{39}$Ar data.

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

## Acknowledgements

We thank the captain and the crew of RV *Meteor* for excellent support, B. Bogner for supporting the sampling at sea and our scientific and technical group, as well as several student assistants for their help with fieldwork and laboratory measurements. We thank S. Beyersdorfer for his contribution to the preparation of the samples, the complete team of the KIP workshop for machining the apparatus, V. Rädle for carefully reading the manuscript, and R. Erven for the graphic design of Figure 1. This research was supported by the Deutsche Forschungsgemeinschaft as part of the Sonderforschungsbereich 754 Climate-Biogeochemistry Interactions in the Tropical Ocean and two joint projects (OB 164/11-1, AE 93/14-1 and OB 164/12-1, AE 93/17-1) as well as by the European Research Commission Advanced Grant EntangleGen (project ID 694561). S.E. gratefully acknowledges support through the scholarship program of the Studienstiftung des deutschen Volkes.

## Author contributions

A.K. and T.T. conducted the sampling, S.E. performed the $^{39}$Ar analyses with assistance from Z.F., L.R. and M.S. T.T. and T.S. analysed the TTD. W.A. and M.K.O. supervised the project. All authors worked on the manuscript.

## Additional information

**Competing interests:** The authors declare no competing interests.

