## [Peer Review File · Nature Communications]

Reviewer #1 (Remarks to the Author):

This manuscript presents new measurements of ^{39}Ar in the ocean, and uses these measurements to constrain the transit time distribution (TTD) and uptake of anthropogenic carbon (Cant). The ability to measure ^{39}Ar with smaller water samples is a major advancement that has, as this paper shows, the ability to improve constraints on both TTDs and Cant. The manuscript is well written and presents novel results that are important and of interest to a wide community. I think it is suitable for publication in Nature Communications after only a few minor revisions. My specific comments are below, and point 1 is probably the most significant (but should be easy to address).

Specific Comments

1. The impact of new constraints on estimates of Cant are mentioned in bold text at front of the manuscript, but only gets a brief mention on lines 165-169 with no details of the calculation shown. More information is needed on how you calculated 6.9 umol/kg , and I think this needs to be shown in a figure. I assume you used the TTD method to estimate Cant, and I think you should show profiles of Cant calculated using standard $\Delta/\Gamma=1$ calculation and your newer ^{39}Ar constrained estimate. This will really highlight the value of these new measurements.
2. Although using the inverse Gaussian approach is OK for this analysis, I think it would be good to mention that the maximum entropy approach (ref 9) is an alternative (and probably better, but more complicated) approach. A sentence in Transit Time Distribution will likely be all that is needed.
3. I don't understand the statement on line 143-144 that upper regime is poorly constrained by ^{39}Ar . In this region doesn't the ^{39}Ar alone gives a constraint on the mean age (i.e. if mean age is much less than decay time of tracer the time inferred just from the tracer is close to the mean age).
4. I find the increase in mean age at the bottom of profile interesting, and wonder if you want to highlight this a little more. Better ventilated flow near the ocean floor may not be a surprise but have previous measurements in the tropics shown this?

Reviewer #2 (Remarks to the Author):

Review of “ ^{39}Ar dating with small samples resolves ocean ventilation” by Ebser et al.

This is a nice short manuscript reporting on important advances. The implications will be of broad interest to the oceanographic and climate communities. The authors report on applying a new technique for measuring Ar-39 in small seawater samples in order to constrain the ocean’s transit-time distribution, and hence its ventilation. Because the decay timescale of Ar-39 matches the ocean’s deep circulation timescales, Ar-39 is an ideal ventilation and circulation tracer. This paper demonstrates the additional transit-time information that Ar-39 concentrations provide within the context of the widely employed inverse-Gaussian (IG) TTD analysis. While this illustration is nice, the important point of the paper is that Ar-39 can now be measured routinely as part of hydrographic surveys because the new optical atom-trap technique requires only 5-liter water samples (200 times less than the previously used decay counting method), thus making it feasible to collect many samples at sea. This certainly has the potential to revolutionize our ability to constrain ocean ventilation and the ocean’s transport generally. I recommend that the manuscript be published after the authors address the issues raised below.

My main criticism of this paper is that it does not adequately discuss the assumptions and limitations of the IG TTD method that is used to illustrate the power of having Ar-39 measurements. While the TTD method is well established and simple to use, it makes several assumptions that may not be applicable everywhere for the profiles analyzed in the manuscript. While the TTD method is fine for illustrating the utility of Ar-39 , the assumptions that are tacitly build into the method should be more clearly articulated, and the unquantified uncertainty stemming from these assumptions should be acknowledged when interpreting the results. There are several key assumptions that should be stated explicitly: (i) equation (2) (please number displayed equations) assumes that ocean transport is steady. The ocean’s real propagator G depends on both transit time τ and on source time $t_s - \tau$ (using t_s as sample time in the notation of the authors). (ii) The real G also depends on source location but this is neglected in Eq. (2), which convolves only over transit time and hence assumes that there is an effective single dominant source region – in reality the properties of the sample are an admixture of water ventilated in different regions that may or may not have the same source concentration history. The IG form for G , which happens to be also the solution of a simple 1-d flow model as the authors mention, tacitly assumes a single dominant source region. (iii) Relatedly, the IG form is unimodal and thus tacitly assumes that there is only a single dominant advective-diffusive transport pathway. This third assumption is particularly relevant here because the deep waters of this study are an admixture of Southern-Ocean-ventilated and North-Atlantic-ventilated waters (relatively long and short transit times, respectively). It is therefore possible that the true TTD [assuming similar input functions in both hemispheres to allow for assumption (ii)] is bimodal, or at least not of IG form (examples of other TTD shapes can be seen in ocean models, e.g., Primeau, JPO, 35, 545-564, 2005 and other data analysis methods). The mean ages and width parameters inferred assuming an IG form could therefore be biased. (Mean age and width are defined in general and are not merely parameters of the IG distribution). This should be briefly discussed and acknowledged.

When discussing their results, the authors should probably also briefly compare with a recent ventilation analysis of GEOTRACES section GA03 (Holzer, Smethie, and Ting, JGR-Oceans, 123, 2332-2352) that traversed the O₂ minimum off Africa that Ebser et al. analyze. While that study did not have the benefit of Ar-39 constraints, it did not use IG TTDs.

A few line-by-line detailed comments:

The title could be better. Other tracers also resolve aspects of ocean ventilation, and conversely Ar-39 does not resolve decadal processes. I think something like “³⁹Ar dating with small samples provides new key constraints on ocean ventilation” would be better.

L11: Ocean ventilation is more broadly defined as the exchange of the interior with the surface ocean, i.e., ventilation involves both propagation from the surface to the interior as well as from the interior back to the surface. I suggest changing “propagate ... to” to exchange ... with”.

L16: It is not true that SF₆, CFCs and ¹⁴C are “not suitable”. I suggest changing “not suitable to resolve the dynamics” to “not ideal for resolving centennial-scale dynamics”

L26: understanding -> ability to quantify

L28: diffusion -> eddy diffusion

L34 implies -> includes (there is more to ventilation than age)

L85-87: “This way ...” Please rephrase for clarity – I found it hard to understand what was being said: 20 liters total, per sample??

L88: 5 to 8 ml Ar per how large a sample? – please clarify in the text

L102-107 Please change “age” to “transit time” and identify the transit time as tau, so that the symbols of Eq. (1) are defined (number equations). (“age” is easily confused with “mean age” or the generally different “tracer age” – here you mean “transit time”.)

L103 “mixing along the flow path” suggest that there is a single path, which is not the case in the 3d ocean. Better to change to something like “mixing and a multitude of advective-eddy-diffusive paths”

L103 “adequate” → “necessary”

L110 briefly discuss the tacit assumptions baked into this IG form as discussed above.

L110 diffusive → eddy-diffusive

L112 delete “an”

L118 discuss that Eq. (2) assumes stationarity of the flow and a single dominant source region. Additionally, I suggest rewording to something like “... corresponds to the tracer concentration during source year $t_s - \tau$ in the dominant surface source region of the water sample.”

L123 “age” → “transit time” (age is often confused with ideal mean age = mean transit time, and distinct from “tracer age”)

L124 As the second ... (“the” missing)

L126 “limit of quantification” → “detection limit”

L136 please add something like the following to your list of uncertainties: “... as well as by the assumption of the propagator G having a simple IG transit-time dependence” – see main criticism above

L139 “regimes based on ventilation” -> “ventilation regimes”

L142 consistent to -> consistent with

L162 “age distribution” is potentially confusing here as this could mean the spatial distribution of the mean age. I suggest changing this to “Once an estimate of the ocean’s surface-to-interior propagator G is known ...”

Reviewer #3 (Remarks to the Author):

This is a very interesting paper which reported ^{39}Ar dating with only 5L of seawater samples. Compared to previous studies (needed 1000L of seawater) with the Low Level Counting (LLC) method this is a very big improvement, which allows large scale applications of ^{39}Ar dating in oceanography.

The ^{39}Ar dating in this paper was performed with a rather new technique called Atom Trap Trace Analysis (ATTA). The part that is related to this technique was clear and well written. The big improvement in reducing sample size came from the recirculation implemented in the measurement. All of the technical details looks sound to me. The paper was well written and should be published.

I just have a few comments and technical questions below.

1)It would be nice to include a table for the ^{39}Ar dating data in a supplement file.

2)The pressure in the MOT chamber seems to be a little high ($1.4\text{E}-7$ mbar). What is the lifetime of the MOT? What is the detection time of the APD? Is the effect of MOT lifetime on the detection efficiency negligible?

3)The detection efficiency is 94%. Is this mainly due to the setting of the threshold?

4)The correction due to memory effect is about 10 atoms for a 20 h measurement. What is the effect on the uncertainty of the measurement due to this correction? Is it included in the estimation of uncertainty for the reported results? For old samples this could be one of the main sources for uncertainties.

Point-by-point response to the reviewers

Our answers are in red.

Reviewer #1 (Remarks to the Author):

This manuscript presents new measurements of ^{39}Ar in the ocean, and uses these measurements to constrain the transit time distribution (TTD) and uptake of anthropogenic carbon (C_{ant}). The ability to measure ^{39}Ar with smaller water samples is a major advancement that has, as this paper shows, the ability to improve constraints on both TTDs and C_{ant} . The manuscript is well written and presents novel results that are important and of interest to a wide community. I think it is suitable for publication in Nature Communications after only a few minor revisions. My specific comments are below, and point 1 is probably the most significant (but should be easy to address).

Specific Comments

1. The impact of new constraints on estimates of C_{ant} are mentioned in bold text at front of the manuscript, but only gets a brief mention on lines 165-169 with no details of the calculation shown. More information is needed on how you calculated 6.9 $\mu\text{mol}/\text{kg}$, and I think this needs to be shown in a figure. I assume you used the TTD method to estimate C_{ant} , and I think you should show profiles of C_{ant} calculated using standard $\Delta/\Gamma=1$ calculation and your newer ^{39}Ar constrained estimate. This will really highlight the value of these new measurements.

We thank the reviewer for this suggestion. We had kept the details of that calculation short deliberately in order not to move attention from the main message that; 1) ^{39}Ar data can constrain ventilation where we previously could not, 2) we are able to measure ^{39}Ar on small volumes, and 3) that we might get surprising results as obtained for the LSW in the tropics.

We have now added some more details in the text and provide an additional figure 4 with a profile of C_{ant} . Additionally we removed the C_{ant} estimates from the abstract.

2. Although using the inverse Gaussian approach is OK for this analysis, I think it would be good to mention that the maximum entropy approach (ref 9) is an alternative (and probably better, but more complicated) approach. A sentence in Transit Time Distribution will likely be all that is needed.

As suggested we added a sentence on the maximum entropy approach and its utility.

3. I don't understand the statement on line 143-144 that upper regime is poorly constrained by ^{39}Ar . In this region doesn't the ^{39}Ar alone gives a constraint on the mean age (i.e. if mean age is much less than decay time of tracer the time inferred just from the tracer is close to the mean age).

We agree that due to the linear behavior of the decay curve for short times, e.g. if the mean is much less than the decay time of the tracer, there is no significant difference between the tracer age and the mean age.

We added some text to explain that the uncertainty in the ^{39}Ar measurements is so that other tracer couples, e.g. SF6 and CFC12, are better suited for these waters. Obviously with an increased precision of the ^{39}Ar measurement, it would indeed be a good tracer for the fast ventilated upper waters as well, for instance combined with SF6 (the concentration of SF6 is still increasing in the atmosphere).

4. I find the increase in mean age at the bottom of profile interesting, and wonder if you want to high this a little more. Better ventilated flow near the ocean floor may not be a surprise but have previous measurements in the tropics shown this?

We thank the reviewer for this suggestion. Although a number of papers have been dealing with the transport of NADW, the increase we see here is coming from the south, i.e. AABW. We don't think this has been described in the literature previously for the tropical Atlantic. We added a sentence to this.

Reviewer #2 (Remarks to the Author):

Review of "39Ar dating with small samples resolves ocean ventilation" by Ebser et al.

This is a nice short manuscript reporting on important advances. The implications will be of broad interest to the oceanographic and climate communities. The authors report on applying a new technique for measuring Ar-39 in small seawater samples in order to constrain the ocean's transit-time distribution, and hence its ventilation. Because the decay timescale of Ar-39 matches the ocean's deep circulation timescales, Ar-39 is an ideal ventilation and circulation tracer. This paper demonstrates the additional transit-time information that Ar-39 concentrations provide within the context of the widely employed inverse-Gaussian (IG) TTD analysis. While this illustration is nice, the important point of the paper is that Ar-39 can now be measured routinely as part of hydrographic surveys because the new optical atom-trap technique requires only 5-liter water samples (200 times less than the previously used decay counting method), thus making it feasible to collect many samples at sea. This certainly has the potential to revolutionize our ability to constrain ocean ventilation and the ocean's transport generally. I recommend that the manuscript be published after the authors address the issues raised below.

We thank the reviewer for the thoughtful, constructive and detailed comments and suggestions to improve the manuscript.

My main criticism of this paper is that it does not adequately discuss the assumptions and limitations of the IG TTD method that is used to illustrate the power of having Ar-39 measurements. While the TTD method is well established and simple to use, it makes several assumptions that may not be applicable everywhere for the profiles analyzed in the manuscript. While the TTD method is fine for illustrating the utility of Ar-39, the assumptions that are tacitly build into the method should be more clearly articulated, and the unquantified uncertainty stemming from these assumptions should be acknowledged when interpreting the results. There are several key assumptions that should be stated explicitly:

- (i) equation (2) (please number displayed equations) assumes that ocean transport is steady. The ocean's real propagator G depends on both transit time τ and on source time $t_s - \tau$ (using t_s as sample time in the notation of the authors).

- (ii) The real G also depends on source location but this is neglected in Eq. (2), which convolves only over transit time and hence assumes that there is an effective single dominant source region – in reality the properties of the sample are an admixture of water ventilated in different regions that may or may not have the same source concentration history. The IG form for G , which happens to be also the solution of a simple 1-d flow model as the authors mention, tacitly assumes a single dominant source region.
- (iii) Relatedly, the IG form is unimodal and thus tacitly assumes that there is only a single dominant advective-diffusive transport pathway. This third assumption is particularly relevant here because the deep waters of this study are an admixture of Southern-Ocean-ventilated and North-Atlantic-ventilated waters (relatively long and short transit times, respectively). It is therefore possible that the true TTD [assuming similar input functions in both hemispheres to allow for assumption (ii)] is bimodal, or at least not of IG form (examples of other TTD shapes can be seen in ocean models, e.g., Primeau, JPO, 35, 545-564, 2005 and other data analysis methods). The mean ages and width parameters inferred assuming an IG form could therefore be biased. (Mean age and width are defined in general and are not merely parameters of the IG distribution). This should be briefly discussed and acknowledged.

We thank the reviewer for the detailed feedback on possible shortcomings in the TTD method. We are aware of the limitations of the IG-TTD approach, but choose not to dwell on those in the manuscript to not distract the reader from the main message (as noted by the reviewer) which is, that we are now able to routinely measure ^{39}Ar and that we use the TTD method to illustrate the potential of the new tracer. We have now added a few sentences on the limitations of the TTD method. It might be possible to resolve several of these limitations with simultaneous observations of additional transient.

When discussing their results, the authors should probably also briefly compare with a recent ventilation analysis of GEOTRACES section GA03 (Holzer, Smethie, and Ting, JGR-Oceans, 123, 2332-2352) that traversed the O2 minimum off Africa that Ebser et al. analyze. While that study did not have the benefit of Ar-39 constraints, it did not use IG TTDs.

We have now included a sentence comparing the results of the two studies, using a different set of transient tracers and methods for evaluating the transient tracer data.

A few line-by-line detailed comments:

The title could be better. Other tracers also resolve aspects of ocean ventilation, and conversely Ar-39 does not resolve decadal processes. I think something like “ ^{39}Ar dating with small samples provides new key constraints on ocean ventilation” would be better.

We thank the reviewer for the suggested title, which we accepted.

L11: Ocean ventilation is more broadly defined as the exchange of the interior with the surface ocean, i.e., ventilation involves both propagation from the surface to the interior as well as from the interior back to the surface. I suggest changing “propagate ... to” to exchange ... with”.

Good suggestion, done.

L16: It is not true that SF6, CFCs and 14C are “not suitable”. I suggest changing “not suitable to resolve the dynamics” to “not ideal for resolving centennial-scale dynamics”

We adapted our sentence: “However, their dating ranges are not ideal to resolve the centennial-dynamics of the deep ocean,...”.

L26: understanding -> ability to quantify

Done

L28: diffusion -> eddy diffusion

Done

L34 implies -> includes (there is more to ventilation than age)

Done

L85-87: “This way ...” Please rephrase for clarity – I found it hard to understand what was being said: 20 liters total, per sample??

We substituted for clarity “this way” by “By combining three Niskin bottles to one sample of ~20 litres...”

L88: 5 to 8 ml Ar per how large a sample? – please clarify in the text

The samples are ~20 liters, but since the solubility of argon (and all other gases) is dependent on temperature and salinity, different amounts of Ar can be extracted from each sample; with the previous adaptation this should be clearer. Additionally we specified the sample size again to ~20 litres.

L102-107 Please change “age” to “transit time” and identify the transit time as tau, so that the symbols of Eq. (1) are defined (number equations). (“age” is easily confused with “mean age” or the generally different “tracer age” – here you mean “transit time”.)

Thanks for that suggestion, corrected.

L103 “mixing along the flow path” suggest that there is a single path, which is not the case in the 3d ocean. Better to change to something like “mixing and a multitude of advective-eddy-diffusive paths”

We changed that part, but left out “eddy” to not exclude dia-pycnal diffusion, which is not so driven by eddy activities.

L103 “adequate” -> “necessary”

Done

L110 briefly discuss the tacit assumptions baked into this IG form as discussed above.

We added a sentence, so that the tacit assumptions are mentioned explicitly now.

L110 diffusive -> eddy-diffusive

Done

L112 delete "an"

Done

L118 discuss that Eq. (2) assumes stationarity of the flow and a single dominant source region. Additionally, I suggest rewording to something like "... corresponds to the tracer concentration during source year $t_s - \tau$ in the dominant surface source region of the water sample."

We added the assumption of a stationary flow explicitly again and edited the sentence as suggested.

L123 "age" -> "transit time" (age is often confused with ideal mean age = mean transit time, and distinct from "tracer age")

Done

L124 As the second ... ("the" missing)

Done

L126 "limit of quantification" -> "detection limit"

We did not apply this change as there is a difference between "detection limit" and "quantification limit", the former often defined as a signal 3-times the background noise, whereas the quantification limit is defined as 7- times the background noise. Although differences in the definition do exist, the use of quantification limit is a stronger statement, which is required, since we need to quantify the concentration of CFC-12. All our data are above the limit of quantification.

L136 please add something like the following to your list of uncertainties: "... as well as by the assumption of the propagator G having a simple IG transit-time dependence" – see main criticism above

Done

L139 "regimes based on ventilation" -> "ventilation regimes"

Done

L142 consistent to -> consistent with

Done

L162 "age distribution" is potentially confusing here as this could mean the spatial distribution of the mean age. I suggest changing this to "Once an estimate of the ocean's surface-to-interior propagator G is known ..."

Good suggestion, but we used the simpler (but we believe still correct) expression “Once the transit time distribution of a water body...”

Reviewer #3 (Remarks to the Author):

This is a very interesting paper which reported ^{39}Ar dating with only 5L of seawater samples. Compared to previous studies (needed 1000L of seawater) with the Low Level Counting (LLC) method this is a very big improvement, which allows large scale applications of ^{39}Ar dating in oceanography.

The ^{39}Ar dating in this paper was performed with a rather new technique called Atom Trap Trace Analysis (ATTA). The part that is related to this technique was clear and well written. The big improvement in reducing sample size came from the recirculation implemented in the measurement. All of the technical details looks sound to me. The paper was well written and should be published.

I just have a few comments and technical questions below.

1) It would be nice to include a table for the ^{39}Ar dating data in a supplement file.

We will provide a table in the supplementary information. In addition, all data can be download in digital format from PANGAEA: Link of the ^{39}Ar -data [<https://doi.pangaea.de/10.1594/PANGAEA.894708>] and of the CTD-data [<https://doi.org/10.1594/PANGAEA.860481>]

2) The pressure in the MOT chamber seems to be a little high ($1.4\text{E}-7$ mbar). What is the lifetime of the MOT? What is the detection time of the APD? Is the effect of MOT lifetime on the detection efficiency negligible?

The pressure in the MOT chamber is measured directly with a hot cathode gauge. The mean lifetime of a single atom is 290 ms, which is consistent with the pressure measurements. The APD is read out with a binning of 1ms. To detect single atoms, the APD-signal has to be filtered. Atoms with a lifetime shorter than 40 ms are difficult to detect and might be missed, which reduces the detection efficiency to 94%. This detection efficiency depends on the mean lifetime. We are comparing the count rate (atoms/h) of the ocean sample with the count rate of the reference sample, so that it is only important that the mean lifetime and thus the detection efficiency is constant during and in between the measurements.

We added these numbers in the method section.

3) The detection efficiency is 94%. Is this mainly due to the setting of the threshold?

See previous answer.

4) The correction due to memory effect is about 10 atoms for a 20 h measurement. What is the effect on the uncertainty of the measurement due to this correction? Is it included in the estimation of uncertainty for the reported results? For old samples this could be one of the main sources for uncertainties.

Yes, these uncertainties are included in our Bayesian analysis and reported in our results. For older water (<10 pmAr) this is the major source of uncertainties. We know the mean value of counted atoms due to the contamination from many measurements with a blank sample very well, but the particular number

of those atoms is unknown for each single measurement and follows the Poisson counting statistic, too. The uncertainty is thus around 10 ± 3 atoms. Even for the oldest sample of this ocean study ($\sim 35 \text{ pmAr}$) we counted on average 54 atoms per measurement, so that the uncertainties due to the memory effect play only a minor role in this study.

We added one sentence to clarify that the uncertainties to the contamination are included.